# Inactivation of Aldehyde Dehydrogenase by Disulfiram in the Presence and Absence of Lipoic Acid or Dihydrolipoic Acid: An in Vitro Study

**DOI:** 10.3390/biom9080375

**Published:** 2019-08-16

**Authors:** Anna Bilska-Wilkosz, Magdalena Górny, Małgorzata Iciek

**Affiliations:** Chair of Medical Biochemistry, Jagiellonian University Medical College, 31-034 Kraków, Poland

**Keywords:** aldehyde dehydrogenase, dihydrolipoic acid, disulfiram, dithiothreitol, lipoic acid

## Abstract

The inhibition of aldehyde dehydrogenase (ALDH) by disulfiram (DSF) in vitro can be prevented and/or reversed by dithiothreitol (DTT), which is a well-known low molecular weight non-physiological redox reagent commonly used in laboratory experiments. These observations inspired us to ask the question whether the inhibition of ALDH by DSF can be preserved or abolished also by dihydrolipoic acid (DHLA), which is the only currently known low molecular weight physiological dithiol in the body of humans and other animals. It can even be metaphorized that DHLA is an “endogenous DTT”. Lipoic acid (LA) is the oxidized form of DHLA. We investigated the inactivation of ALDH derived from yeast and rat liver by DSF in the presence or absence of LA or DHLA. The results clearly show that DHLA is able both to restore and protect ALDH activity blocked by DSF. The proposed mechanism is discussed.

## 1. Introduction

The aldehyde dehydrogenase superfamily (ALDHs) is composed of NAD(P)^+^-dependent enzymes. The ALDH proteins were found to be present in all subcellular compartments: cytosol, mitochondria, endoplasmic reticulum, and nucleus. Members of the aldehyde dehydrogenase gene (ALDH) superfamily play an important role in the enzymatic detoxification of endogenous and exogenous aldehydes and in the formation of molecules that are important in cellular processes, like retinoic acid, betaine, and gamma-aminobutyric acid. However, acetaldehyde (CH_3_CHO), produced mostly by oxidation of ingested ethanol (C_2_H_5_OH, EtOH) is the best-known substrate of ALDH. 

Disulfiram (tetraethylthiuram disulfide, tetraethylthioperoxydicarbonic diamide, DSF) commercially known as Antabuse, has been used since 1948 as an alcohol-aversive agent in the treatment of alcoholism [1]. By inhibiting ALDH activity, DSF causes EtOH intolerance due to poisoning with acetaldehyde, the concentration of which rises abruptly after EtOH consumption when ADLH had been inhibited. It causes a spectrum of undesirable side-effects, including vertigo, nausea, facial flushing, low blood pressure, tachycardia, and in extreme cases, even death. The inhibition was found to be irreversible, and restoration of ALDH activity was dependent on protein synthesis [2]. It indicates that at present there is no drug able to efficiently abolish these symptoms, collectively called a disulfiram ethanol reaction (DSF-EtOH reaction, DER). 

Moreover, it should be remembered that, apart from DSF, many other drugs, i.e., antibiotics are implicated in inducing a DSF-like reaction [3,4,5]. Furthermore, DSF-like side effects can occur when EtOH is used together with some plant-derived pharmaceuticals or dietary supplements since they inhibit the activity of ALDH [6,7]. Also, it is worth mentioning that there are certain food ingredients, like coprine, daidzin, or garlic-derived sulfur compounds, which can produce DSF-like reactions [1,8].

In accordance with literature data, inhibition of ALDH by DSF in vitro can be prevented and/or reversed by dithiothreitol ((2S,3S)-1,4-dimercaptobutane-2,3-diol; DTT), which is a well-known low molecular weight non-physiological redox reagent commonly used in laboratory experiments [9,10]. It inspired us to ask the question whether ALDH inhibition by DSF in vitro can be preserved or abolished also by dihydrolipoic acid (6,8-dimercaptooctanoic acid, DHLA) and/or lipoic acid (5-[(3R)-dithiolan-3-yl] pentanoic acid, LA). Why DHLA? Because DHLA is a low-molecular weight reductant which, like DTT, has two thiol groups per molecule. DHLA is the reduced form of LA. Lipoic acid is synthesized in mammals in the mitochondria of the liver and other tissues. Most LA/DHLA in the body is found tightly bound to proteins by an amide linkage with a lysine residue. This lipoamide/dihydrolipoamide bond functions mainly as a cofactor of multienzymatic mitochondrial complexes catalyzing oxidative decarboxylation of α-ketoacids and the glycine cleavage system. It should be remembered that DHLA is the only currently known low molecular weight physiological dithiol in the body of humans and other animals. It can be metaphorized that DHLA is an “endogenous DTT”. 

Apart from endogenous synthesis, LA is also absorbed from food. LA has been detected in various natural products. The exogenously administered LA is also successfully used as a drug in the treatment of diabetic polyneuropathy [11], liver cirrhosis, and heavy metal intoxication [12,13]. Recent experimental and clinical studies have proven a beneficial effect of LA also in kidney diseases [14], heart diseases [15], Parkinson’s disease [16], inflammation [17,18], retinal diseases [19], and even cyanate intoxication [20]. Moreover, some animal or human studies have shown a significant increase in the activity and expression of ALDH2 under the influence of LA [21,22,23]. It suggests that beneficial action of LA in some cases is connected with up-regulating ALDH2 activity.

It should be underlined that commercially available preparations (drugs, dietary supplements) mostly contain the oxidized form (LA) since the reduced form (DHLA) is very unstable. It is assumed that after administration, LA is transported to the intracellular compartments and is reduced to DHLA in the reaction catalyzed by NAD(P)H-dependent enzymes (dihydrolipoate dehydrogenase, glutathione reductase, and thioredoxin reductase) [24]. It is suggested that therapeutic effects of LA supplementation can be attributed mainly to the potent antioxidant properties of DHLA.

Hence, considering a very wide spectrum of pharmacological activities of LA/DHLA and the fact that both DTT and DHLA contain two thiol groups, which may suggest their similar redox properties, we hypothesized that LA/DHLA can reactivate ALDH activity inhibited by DSF. 

Thus, for the first time, in an in vitro study, we investigated the inactivation of ALDH by DSF in the presence and absence of LA and DHLA. Due to high homology between human and yeast ALDH, we used the latter in our study. The analogical study using rat liver-derived ALDH was performed. In the future we plan to carry out such studies in vivo on laboratory animals. 

## 2. Materials and Methods

### 2.1. Chemicals

In this study, the formulation Thiogamma was used, which contains LA as the pharmacologically active substance. Thiogamma was obtained from Hexal AG, (Holzkirchen, Germany). It contains LA in the form of meglumine salt and is used medically for intravenous infusion. DHLA was obtained from Sigma-Aldrich Chemical Company (Poznań, Poland) as a liquid. Purified yeast ALDH as well as DSF, NAD, and propionaldehyde were provided by Sigma-Aldrich Chemical Company. All other reagents were of analytical grade and were obtained from the Polish Chemical Reagent Company (POCh, Gliwice, Poland). DSF was dissolved in ethanol and then after dilution with buffer was used for the experiment. The same concentration of ethanol was added to control samples.

### 2.2. Homogenate Preparation

The liver from one male Wistar rat was used to obtain a homogenate. This animal was chosen from a control group of another experiment which was approved by the local ethics committee for Animal Research in Krakow (number registration 94/VIII/2011). The frozen liver was weighed and immediately homogenized (1 g of tissue in 4 mL of 0.1 M phosphate buffer, pH 7.4) using an IKA-ULTRA TURRAX T8 homogenizer (IKA Poland Sp. z o. o. Company, Warsaw, Poland). Homogenate was kept in ice and was used immediately for experiments.

### 2.3. Determination of Yeast ALDH Activity

The final concentration of ALDH was 0.3 mg/mL in all experiments. Three control samples were prepared: control 1—containing only ALDH, control 2—containing ALDH with 1 mM LA, and control 3—containing ALDH and 1 mM DHLA. All samples were dissolved in 50 mM sodium phosphate buffer (pH 8.0). The mixtures containing ALDH, DSF, and LA/DHLA were made simultaneously (in detail in Section 2.5). The control samples and all test mixtures were incubated at a temperature of 25 °C. Data were calculated as a percentage relative to control 1 (ALDH alone). The activity of yeast ALDH was assayed as described previously [8,25]. 

Briefly, 710 µL of 50 mM sodium phosphate buffer (pH 8.2), 200 µL of 5 mM NAD, and 40 µL of 1 mM propionaldehyde were pipetted into a cuvette. The reaction was initiated by the addition of 50 µL of the indicated control sample or mixture to the cuvette and absorbance change at 340 nm was monitored for 2 min at 25 °C to calculate the rate of NADH production and to compare the data to the sample containing only ALDH without DSF (control sample). Data are presented as a percentage relative to control (100%). 

### 2.4. Determination of ALDH Activity in the Rat Liver Homogenate

The assay mixture contained liver homogenate, sodium phosphate buffer (pH 8.2), NAD, EDTA, 4-methylpyrazole, and rotenone. The reaction was initiated by the addition of propionaldehyde as a substrate. 4-Methylpyrazole was added to inhibit alcohol dehydrogenase, and rotenone to inhibit mitochondrial NADH oxidase. The blank sample in which the homogenate was omitted was run simultaneously. The activity of ALDH was calculated using the molar extinction coefficient of NADH of 6.22 mM^−1^ cm^−1^ at 340 nm with the use of a modified protocol published earlier [26]. Specific activity of the enzyme was expressed as nanomoles of NADH produced per 1 mg of protein per 1 min. The protein content was measured using the method of Lowry et al. [27]. 

### 2.5. Series of Mixtures

#### 2.5.1. Assay Using Yeast ALDH

Three series of mixtures were prepared, which is explained in detail in Scheme 1, Scheme 2, and Scheme 3. Series I was prepared as follows:Mixture 1. ALDH was incubated with 0.1 mM DSF in 50 mM sodium phosphate buffer for 20 min.Mixture 2. ALDH was preincubated with 0.1 mM DSF in 50 mM sodium phosphate buffer for 5 min, then LA was added (final concentration 1 mM) and the mixture was incubated for a further 15 min.Mixture 3. ALDH was preincubated with 0.1 mM DSF in 50 mM sodium phosphate buffer for 5 min, then DHLA was added (final concentration 1 mM) and the mixture was incubated further for 15 min.

Series II was prepared as follows:Mixture 1. ALDH was preincubated alone for 15 min, then 0.1 mM DSF and the mixture was incubated for a further 5 min.Mixture 2. ALDH was preincubated with 1mM LA in 50 mM sodium phosphate buffer for 15 min, then 0.1 mM DSF was added and the mixture was incubated for a further 5 min.Mixture 3. ALDH was preincubated with 1 mM DHLA in 50 mM sodium phosphate buffer for 15 min, then 0.1 mM DSF was added and the mixture was incubated for a further 5 min.

Series III was prepared as follows:
Mixture 1. DSF (0.1 mM) in 50 mM sodium phosphate buffer was preincubated for 15 min, then ALDH was added and the mixture was incubated for a further 5 min. Mixture 2. DSF (0.1 mM) in 50 mM sodium phosphate buffer was preincubated with 1 mM LA for 15 min, then ALDH was added and the mixture was incubated for a further 5 min.Mixture 3. DSF (0.1 mM) in 50 mM sodium phosphate buffer was preincubated with 1 mM DHLA for 15 min, then ALDH was added and the mixture was incubated for a further 5 min.

#### 2.5.2. Assay Using Rat Liver Homogenate-Derived ALDH

The same experiment as described above was performed using rat liver homogenate as a source of ALDH. Three series of mixtures were prepared correspondingly to previously presented Scheme 1, Scheme 2 and Scheme 3, where rat homogenate (diluted 10 times with 50 mM phosphate buffer, pH 8.2) was added instead of yeast ALDH.

### 2.6. Statistical Analysis

All statistical calculations were carried out with the STATISTICA 10.0 computer program (Statsoft Inc., Tulsa, OK, USA). Data are presented as the mean ± standard deviation (SD) of three independent experiments in triplicates. The statistical calculations were performed using a one-way analysis of variance (ANOVA), and post-hoc Tukey test. For all data, the values of *p* < 0.05 were considered to be statistically significant.

## 3. Results 

### 3.1. Yeast ALDH

#### 3.1.1. Series I

The experiment indicated that 20 min incubation with DSF induced a decrease in the ALDH activity to 14.6% (mixture 1) as compared to the control (100%). A 5 min preincubation of the enzyme with 0.1 mM DSF and then a 15 min incubation with 1 mM LA or DHLA (mixture 2 and 3, respectively) caused a decrease in ALDH activity to 17.7% and 41.4%, respectively, compared with the control (100%). In relation to DSF, this result was significant for DHLA but not for LA, which indicated that LA was not able to restore ALDH activity blocked by DSF. The results obtained in Series I are presented in Figure 1.

#### 3.1.2. Series II

The experiment demonstrated that a 5 min incubation with DSF lowered ALDH activity to 33% (mixture 1) compared to the control (100%). When the samples of ALDH were preincubated with 1 mM LA or DHLA (mixture 2 and 3, respectively) for 15 min and then with 0.1 mM DSF for 5 min, the ALDH activity changed to 38.2% and 134.5%, respectively, compared to the control (100%). In relation to DSF, this result was significant for DHLA but not for LA, which indicated that LA was not able to protect ALDH against the inhibitory effect of DSF. However, in the presence of DHLA, activity of the enzyme not only returned to the control value but exceeded it markedly (134.5%). It means that DHLA was able to protect ALDH against the inhibitory effect of DSF. The results obtained from this experiment in Series II are presented in Figure 2. 

#### 3.1.3. Series III

Series III was performed according to Scheme 3 in order to confirm that DSF directly interacts with DHLA. In this experiment DSF was preincubated with LA/DHLA and then ALDH was added. DSF alone (mixture 1) decreased the activity of ALDH to 7.0% compared to the control (100%). A 15-min preincubation of 1 mM LA with 0.1 mM DSF and then 5 min incubation with ALDH (mixture 2) caused a decrease in the enzyme activity to about 1% as compared to the control (100%). Analogical experiment with 0.1 mM DSF and 1 mM DHLA (mixture 3) revealed that DHLA protected ALDH activity against the inhibitory effect of DSF. The results obtained from this experiment in Series III are presented in Figure 3. 

### 3.2. Rat Liver-Derived ALDH 

#### 3.2.1. Series I

The experiment performed according to Scheme 1 with rat liver homogenate indicated that 20 min incubation with DSF (mixture 1) induced a decrease in the ALDH activity to 31.8% as compared to the control (100%). A five-minute preincubation of the homogenate with 0.1 mM DSF and then a 15 min incubation with 1 mM LA (mixture 2) caused a decrease in ALDH activity to 33.2% compared with the control (100%). On the other hand, 1 mM DHLA (mixture 3) restored ALDH activity compared to DSF-inhibited enzyme (mixture 1). The results obtained from the study using rat liver homogenate in Series I are presented in Figure 4. 

#### 3.2.2. Series II

The study performed according to Scheme 2 with rat liver homogenate indicated that 5 min incubation with DSF (mixture 1) induced a decrease in the ALDH activity to 29.6% as compared to the control (100%). A 15 min preincubation of the homogenate with 1 mM LA and next 5 min incubation with 0.1 mM DSF (mixture 2) caused a decrease in ALDH activity to 17.1% compared with the control (100%). On the other hand, 1 mM DHLA (mixture 3) restored ALDH activity compared to DSF-inhibited enzyme (mixture 1). The results obtained from the study using rat liver homogenate in Series II are presented in Figure 5. 

#### 3.2.3. Series III

Series III was performed with rat liver homogenate according to Scheme 3. In this experiment DSF was preincubated with LA/DHLA and then rat liver homogenate was added. DSF alone (mixture 1) decreased the activity of rat liver-derived ALDH to 31.7% compared to the control (100%). A 15 min preincubation of 1 mM LA with 0.1 mM DSF and then 5 min incubation with rat liver-derived ALDH (mixture 2) caused a decrease in the enzyme activity to 61.8% as compared to the control (100%). An analogical experiment with 0.1 mM DSF and 1 mM DHLA (mixture 3) revealed that DHLA protected ALDH activity against the inhibitory effect of DSF. The results obtained from the study using rat liver homogenate in Series III are presented in Figure 6. 

## 4. Discussion

In line with literature data, the present studies demonstrated that DSF inhibited ALDH activity in vitro. Kitson proposed that DSF inhibited ALDH in vitro by forming an intermolecular mixed disulfide with the thiol group in the enzyme’s active site [2]. The mechanism of interaction between DSF and ALDH is analogous to the commonly known reaction between thiols and Ellman’s reagent [5,5^‘^-dithiobis-(2-nitrobenzoic acid)]. It means that one of the -SH groups of ALDH forms an intermolecular mixed disulfide with diethyldithiocarbamate (DDC) which is a DSF monomer. In other words, ALDH reduces DSF to DDC and simultaneously is oxidized to disulfide. This manner of reduction and oxidation of -SH/SS- (also in vivo), unique for thiol compounds, is called the thiol-disulfide interchange reaction. In this process, a thiol (RSH) reacts with a disulfide (R’SSR’) forming a new disulfide (RSSR’) and a thiol (R’SH) derived from the original disulfide [28,29]. Thus, the possible mechanism of interaction between DSF and ALDH can be expressed as Equation (1):

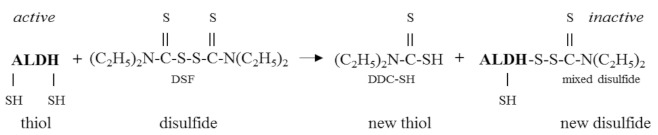
(1)

Hence, it can be stated that ALDH participates here in the carbamoylation reaction. The creation of this inactive carbamoyl protein adduct (ALDH-DDC) finally leads to the formation of an intramolecular disulfide bond between the enzyme’s active site thiol and the thiol of another cysteine residue in the molecule of yeast ALDH [3,30] that can be expressed as Equation (2):
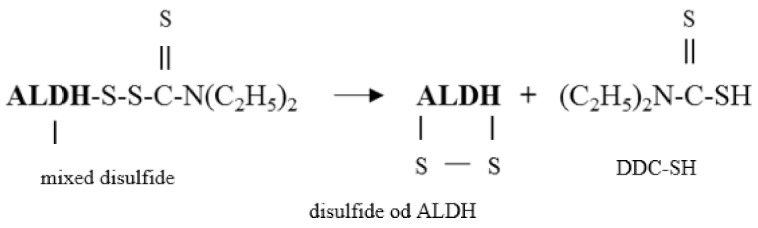
(2)

Several studies demonstrated that DSF could be reduced to DDC also by low-molecular weight thiols (LMWTs), such as DTT, reduced glutathione (GSH), and 2-mercaptoethanol (2-hydroxy-1-ethanethiol; 2-ME) with concomitant oxidation of LMWTs to low-molecular weight disulfides (LMWDs) [2,31,32] that can be expressed as Equation (3):
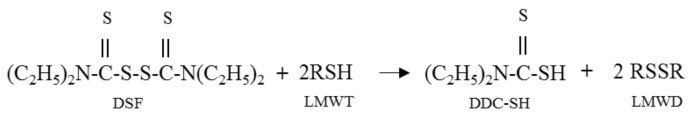
(3)

DDC formed in this reaction in vitro does not inhibit ALDH activity [1]. Thus, in the mixture containing ALDH, DSF, and one of the LMWTs, the two reactions presented above (Equation (1) and Equation (3)) will compete with each other. Kitson showed that LMWTs were able to protect ALDH activity against the inhibitory effect of DSF because DSF was much more efficiently destroyed by LMWTs than by ALDH [33].

The present study demonstrated that DHLA could protect ALDH against the inhibitory effect of DSF. Thus, it is a plausible proposal that DHLA, just like other LMWTs studied in this context, reduces DSF to DDC, which makes a DSF–ALDH interaction impossible, leading to the formation of intermolecular mixed disulfide with the thiol groups of ALDH and thus eliminating the inactivating effect of DSF on this enzyme. This thesis seems to be confirmed by the results obtained in Series III from studies on both yeast ALDH and enzyme derived from rat liver homogenate, as shown in Figure 3 and Figure 6. 

The present study also indicates that inhibition of ALDH by DSF can be reversed by DHLA. Interestingly, DHLA increased ALDH activity by 34% when compared to the control. It suggests that commercially available enzyme preparation contains some oxidized sulfhydryl groups. DHLA acts as a reductant and in this way increases the total ALDH activity. LA as a disulfide had no effect on the activity of ALDH in this experiment. Similar results indicating an increase in ALDH activity under the influence of various concentrations of DHLA were obtained by us earlier (unpublished data). Some authors indicated that the mixed disulfide (ALDH-DDC) (Equation (1)) was only an unstable intermediate product finally leading to formation of an intramolecular disulfide bond between the thiols of cysteine residues in the ALDH molecule [9,34] that can be expressed as Equation (4):

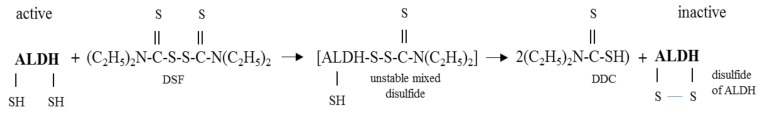
(4)

The proposed mechanism of interaction between ALDH, DSF, and DHLA is shown in Figure 7. It is believed that disulfide bonds in proteins can be reduced only by dithiol compounds [29,35]. However, results of studies are not so unequivocal in the case of ALDH. Vallari and Pietruszko [34] demonstrated in vitro that ALDH activity inhibited by DSF was restored by 2-ME, which suggested that disulfide bonds in ALDH could be reduced also by monothiol compounds. On the other hand, the same authors demonstrated that GSH was not able to reverse of DSF-inhibited ALDH activity [34]. Results on DTT are also ambiguous. While Li and Vallee [36] reported that ALDH inhibition by DSF could be prevented but not reversed by DTT, data from other authors indicated that DTT was able both to restore and protect ALDH activity blocked by DSF in vitro [9,10]. Our previous study revealed that activity of ALDH inhibited by some other reactive sulfur species could be fully restored by DTT, then DHLA, whereas only partially by GSH [8].

## 5. Conclusions

Taken together, our in vitro results clearly show that DHLA but not LA is able both to restore and protect ALDH activity blocked by DSF in vitro. This confirms potential new pharmacological properties of LA/DHLA. It means that LA could find clinical use as a drug that restores the activity of ALDH inhibited by DSF. On the other hand, taking into account that this type of research was carried out for the first time with the use of LA/DHLA, it is obvious that this problem requires further research. In our opinion in vivo studies are needed, which will certainly significantly deepen our knowledge of the relationship between aldehyde dehydrogenase, disulfiram, lipoic acid, and/or dihydrolipoic acid.

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
