# Peer review of "Inactivation of Aldehyde Dehydrogenase by Disulfiram in the Presence and Absence of Lipoic Acid or Dihydrolipoic Acid: An in Vitro Study"

_biomolecules, 2019, doi:10.3390/biom9080375_

Round 1

Reviewer 1 Report

In this manuscript, Bilska-Wilkosz et al. performed a simple in vitro study and demonstrated that DHLA but not LA is able to restore and protect ALDH activity blocked by DSF. The experimental design was poor and the conclusion was not strongly supported by the obtained data.

In the enzymatic assay, why 1 mM LA/DHLA was chose? A dose effect of LA/DHLA should be tested. What is the dose of yeast ALDH protein?

A rationale to use yeast ALDH in this manuscript needs to be provided. A more detailed description on how to measure ALDH activity should be expanded with literature support.

In Figure 1A and 1B, the control group was missing. Why name Figure 1A and 1B? It is suggested to name as Figure 1 and Figure 2.

The experiment needs to be validated in the presence of cultured cells or tissue lysates.

Other thiol-containing compounds are also suggested to test in this manuscript to find evidence as the authors proposed in the discussion section.

Author Response

We are grateful to the Reviewer for his constructive and precise comments on our present manuscript, with which we largely agree.

Please, find below our replies.

- In the enzymatic assay, why 1 mM LA/DHLA was chose? A dose effect of LA/DHLA should be tested. What is the dose of yeast ALDH protein?

The concentration of yeast ALDH was 0.3 mg of protein in all experimental mixtures (it is mentioned in Materials and Methods section).

The concentration of 1 mM LA/DHLA has been chosen based on the literature data. It is one of the most frequently used LA concentrations in in vitro studies (For a few examples, please see the references below)

- 0.4-4 mM LA

Kim KH, Lee B, Kim YR, Kim MA, Ryu N, Jung DJ, Kim UK, Baek JI, Lee KY. Evaluating protective and therapeutic effects of alpha-lipoic acid on cisplatin-induced ototoxicity. Cell Death Dis. 2018 Aug 1;9(8):827.

- 2 mM LA

Yang L, Wang X, Xu J, Wen Y, Zhang M, Lu J, Wang R, Sun X. Integrated transcriptomic and proteomic analyses reveal É‘-lipoic acid-regulated cell proliferation via Grb2-mediated signalling in hepatic cancer cells. J Cell Mol Med. 2018 Jun;22(6):2981-2992.

 - 1-5 mM LA

Ghelani H, Razmovski-Naumovski V, Pragada RR, Nammi S. Attenuation of Glucose-Induced Myoglobin Glycation and the Formation of Advanced Glycation End Products (AGEs) by (R)-α-Lipoic Acid In Vitro. Biomolecules. 2018 Feb 8;8(1). pii: E9.

Moreover, the dose-dependent effect of LA/DHLA was studied by us earlier (data presented during XLV Polish Annual Conference on Catalysis - 2013). Those studies revealed no significant differences in the effect of various tested concentrations (0.5 – 2 mM) of LA/DHLA on ALDH activity. On this basis, the concentration of 1 mM was chosen for further studies.

- A rationale to use yeast ALDH in this manuscript needs to be provided. A more detailed description on how to measure ALDH activity should be expanded with literature support.

We used the yeast ALDH in our study due to high homology between human and yeast ALDH. This clarification has been added in the introduction. Description of the ALDH activity assay method has been enriched in more details and references have been added.

- In Figure 1A and 1B, the control group was missing. Why name Figure 1A and 1B? It is suggested to name as Figure 1 and Figure 2.

The control group has been added to both Figures. Moreover, the names Figure 1 and Figure 2 have been introduced.

-The experiment needs to be validated in the presence of cultured cells or tissue lysates.

Analogical experiments were performed in the presence of rat liver homogenate and obtained results were added to manuscript.

Other thiol-containing compounds are also suggested to test in this manuscript to find evidence as the authors proposed in the discussion section.

As mentioned in the discussion, it seems that disulfide bond in the inactive form of ALDH can be reduced only by compounds containing two sulfhydryl groups (dithiols). To our knowledge, there are no such low molecular compounds apart from DTT and DHLA. Some authors suggested that activity of ALDH inhibited by DSF can be restored by thiol 2-mercaptoethanol [Ref. 34]. On the other hand, the same authors demonstrated that the monothiol GSH was not able to reverse the DSF-induced inhibition of ALDH activity [Ref. 34]. Our previous study revealed that activity of ALDH inhibited by some reactive sulfur species could be fully restored by DTT, then DHLA, whereas only partially by GSH [Ref. 8].

Reviewer 2 Report

Authors demonstrated protective effects of DHLA against DSF inhibitory activity toward ALDH. Although DHLA is well known to protect ALDH2 from inactivation in oxidative stress condition, its effect on ALDH inactivation by DSF are poorly understood. To my knowledge, current work is the first attempt to investigate the effect of interaction between DHLA and DSF on ALDH activity. 

The introduction and discussion are clear and well structured. However, the results and method sections require improvements. 

Authors claim DHLA to directly interact with DSF forming a disulfide bond, thus preventing DSF to bind ALDH. However, it is not directly shown by experiment. It's not clear whether co-incubation of DHLA and DSF results in adduct formation or DHLA is able to reduce disulfide bond between DSF and ALDH. I would recommend authors to introduce another 2 reaction controls: pre-incubate 1) DSF and DHLA, 2) DSF and LA for 15 minutes, then add it to ALDH, incubate for 20 minutes and measure ALDH activity. If ALDH activity will stay the same as in sequential incubations it could be a more clear confirmation that DSF directly interacts with DHLA. 

In addition:

1) DFS needs to be corrected to DSF in Figure 1A 

2) DIS needs to be corrected to DSF in Scheme-Series I, enzyme mixture 3

3) please correct 'contain' to 'mostly contain' at line 71 since there is stabilized DHLA available on the market

4) LA, DHLA, and DSF are poorly dissolved in water, please add information into Method section about solvents used to dissolve all of the compounds (if any). Also please add information whether drug solvents were added to control reactions or not. 

5) please add a Table containing absolute values for reaction rates of all control and enzymatic reactions.  

6) DHLA previously was shown to both restore ALDH2 activity under oxidative stress in vivo and in vitro and increase expression of ALDH2 in various mammalian tissues (see references below). It is relevant to this research and will be a valuable addition to the introduction. 

Wenzel P, et al. Role of reduced lipoic acid in the redox regulation of mitochondrial aldehyde dehydrogenase (ALDH-2) activity. Implications for mitochondrial oxidative stress and nitrate tolerance. J Biol Chem. 2007 Jan 5;282(1):792-9.

Shindyapina AV, et al. The Antioxidant Cofactor Alpha-Lipoic Acid May Control Endogenous Formaldehyde Metabolism in Mammals. Front Neurosci. 2017 Dec 1;11:651. doi: 10.3389/fnins.2017.00651. eCollection 2017.

Li RJ, et al. Alpha-lipoic acid ameliorates oxidative stress by increasing aldehyde dehydrogenase-2 activity in patients with acute coronary syndrome. Tohoku J Exp Med. 2013 Jan;229(1):45-51.

Li JH, et al. Lipoic acid protects gastric mucosa from ethanol-induced injury in rat through a mechanism involving aldehyde dehydrogenase 2 activation. Alcohol. 2016 Nov;56:21-28.

7) Interestingly, DHLA followed by DSF incubation increase ALDH activity by 34% compared to DHLA only (FIgure 1B). Please discuss this unexpected effect.

8) please add the replicate number to Figure descriptions. 

Author Response

We are grateful to the Reviewer for his constructive and precise comments on our present manuscript, with which we largely agree.

Please, find below our replies.

Authors claim DHLA to directly interact with DSF forming a disulfide bond, thus preventing DSF to bind ALDH. However, it is not directly shown by experiment. It's not clear whether co-incubation of DHLA and DSF results in adduct formation or DHLA is able to reduce disulfide bond between DSF and ALDH. I would recommend authors to introduce another 2 reaction controls: pre-incubate 1) DSF and DHLA, 2) DSF and LA for 15 minutes, then add it to ALDH, incubate for 20 minutes and measure ALDH activity. If ALDH activity will stay the same as in sequential incubations it could be a more clear confirmation that DSF directly interacts with DHLA.

We thank the Reviewer for this right and helpful suggestion. We have performed this experiment and the obtained results confirmed our thesis.

In addition:

1) DFS needs to be corrected to DSF in Figure 1A

It has been corrected.

2) DIS needs to be corrected to DSF in Scheme-Series I, enzyme mixture 3

It has been corrected.

3) please correct 'contain' to 'mostly contain' at line 71 since there is stabilized DHLA available on the market

It has been corrected.

4) LA, DHLA, and DSF are poorly dissolved in water, please add information into Method section about solvents used to dissolve all of the compounds (if any). Also please add information whether drug solvents were added to control reactions or not.

In the case of LA, in this study the formulation Thiogamma (from Hexal®AG, Holzkirchen, Germany) was used. It contains LA in the form of meglumine salt and is used medically for intravenous infusion. DHLA was obtained from Sigma-Aldrich Chemical Company as a liquid.

Both substances were then diluted with buffer and used for experiments. DSF was dissolved in ethanol and was used for experiments after dilution with buffer.

We fully agree with Reviewer that control mixtures should contain exactly the same solvents as drug solvents, however in the case of commercially available LA and DHLA it was not possible due to the lack of information concerning details of these solvents. Corresponding concentration of ethanol was added to control samples.

5) please add a Table containing absolute values for reaction rates of all control and enzymatic reactions.

The commercially available yeast ALDH is a highly unstable enzyme. The activity of enzyme in each bottle is different and this is why we have presented the results as a percentage of control instead of absolute values. In each series all controls and enzyme mixtures were performed in triplicates in one experiment. The experiments were then repeated 3 times with another enzyme bottle.

6) DHLA previously was shown to both restore ALDH2 activity under oxidative stress in vivo and in vitro and increase expression of ALDH2 in various mammalian tissues (see references below). It is relevant to this research and will be a valuable addition to the introduction.

Wenzel P, et al. Role of reduced lipoic acid in the redox regulation of mitochondrial aldehyde dehydrogenase (ALDH-2) activity. Implications for mitochondrial oxidative stress and nitrate tolerance. J Biol Chem. 2007 Jan 5;282(1):792-9.

Shindyapina AV, et al. The Antioxidant Cofactor Alpha-Lipoic Acid May Control Endogenous Formaldehyde Metabolism in Mammals. Front Neurosci. 2017 Dec 1;11:651. doi: 10.3389/fnins.2017.00651. eCollection 2017.

Li RJ, et al. Alpha-lipoic acid ameliorates oxidative stress by increasing aldehyde dehydrogenase-2 activity in patients with acute coronary syndrome. Tohoku J Exp Med. 2013 Jan;229(1):45-51.

Li JH, et al. Lipoic acid protects gastric mucosa from ethanol-induced injury in rat through a mechanism involving aldehyde dehydrogenase 2 activation. Alcohol. 2016 Nov;56:21-28.

 Thank you for this suggestion. We have added this information and adequate references to the introduction.

7) Interestingly, DHLA followed by DSF incubation increase ALDH activity by 34% compared to DHLA only (Figure 1B). Please discuss this unexpected effect.

The previous Scheme 2 presenting series II wrongly suggested that ALDH activity after incubation with DHLA and then with DSF was increased compared to ALDH with DHLA only. Both schemes (1 and 2) have been modified. Appropriate control was performed for each enzyme mixture, however, the results in the Figures were calculated in relation to control containing only ALDH (100 %). So, incubation of ALDH with DHLA and then with DSF increased the ALDH activity by 34% compared to the control ALDH. It was connected with the fact that commercially available enzyme contains some oxidized sulfhydryl groups. DHLA acts as a reductant and in this way increases total ALDH activity. LA as a disulfide had no effect on the activity of ALDH in this experiment. Similar results indicating an increase in ALDH activity under the influence of various concentrations of DHLA were obtained by us earlier (data presented during XLV Polish Annual Conference on Catalysis - 2013). This information has been discussed.

8) please add the replicate number to Figure descriptions.

It has been added.

Reviewer 3 Report

The authors point out that the inhibition of aldehyde dehydrogebase (ALDH) by disulfiram (DSF) in vitro can be prevented and/or reversed by dithiothreitol (DTT), which is a well - known low molecular weight non - physiological dithiol. This fact inspired the authors to ask the question whether the inhibition of ALDH by DSF can be preserved or abolished also by dihydrolipoic acid (DHLA), which is the only currently known natural low molecular weight physiological dithiol in the body of humans and animals. The subject addressed by the authors is interesting because it is important from the point of view of basic research.

However, there are many points to be reconsidered.

The authors wrote that the inspiration for the undertaken research is the fact that the inhibition of ALDH by DSF in vitro can be prevented and/or reversed by dithiothreitol (DTT) which is a well - known low molecular weight non - physiological dithiol. The authors compare the structure of DTT with the structure of DHLA, which, like DTT, is a dithiol and unlike DTT, is a physiologically occuring compound in animal organisms. However, the authors also used LA in their work, which is an oxidized form of DHLA and is not a dithiol. If DHLA vs. DTT are compared, LA would have to be compared with the oxidized form of DTT, i.e. trans-4-5- dihydroxy-1,2-dithiane. The authors may discuss this issue, e.g. in the Discussion.

In the Introduction, the authors wrote that "Furthermore, DSF-like side effects can occur when EtOH is used together with some plant-derived pharmaceuticals or dietary supplements which are often used without physicians supervision". In my opinion, specific examples of plant-derived pharmaceuticals or dietary supplements should be given here, as in the next sentence, where the authors present coprine, as an example of compounds which interact with EtOH to produce DSF-like reaction.

I suggest you re-edit the Materials and Methods section. From the reader's point of view, this part is sometimes not very clear and chaotic. Although the reviewer understood this, it took him too much time. It should be corrected.

In the Results, the authors wrote: The obtained results demonstrated that the presence of 1 mM thiols (LA and DHLA) had no effect on the activity of intact ALDH. This sentence should be rewritten because LA is not a thiol. It is a disulfide.

The obtained Results showed that DHLA was able to protect ALDH against the inhibitory effect of DSF (Fig. 1B). The ALDH activity changes here to 134.5% compared to the control (100%). Can the authors explain this? It would be good to discuss this problem in the Discussion.    

I also found a few language errors: typos, punctuation, etc. This should be corrected. For example: page 7 (167) - there are commas instead of dots; page 5 (128), 8 (182) and the following pages: the "in vitro" expression should be written in italics; page 8 (204) and 9 (231) - use the same abbreviation: either Me or 2-Me.           

Author Response

We are grateful to the Reviewer for his constructive and precise comments on our present manuscript with which largely agree.

Please, find below our replies.

-The authors wrote that the inspiration for the undertaken research is the fact that the inhibition of ALDH by DSF in vitro can be prevented and/or reversed by dithiothreitol (DTT) which is a well - known low molecular weight non - physiological dithiol. The authors compare the structure of DTT with the structure of DHLA, which, like DTT, is a dithiol and unlike DTT, is a physiologically occuring compound in animal organisms. However, the authors also used LA in their work, which is an oxidized form of DHLA and is not a dithiol. If DHLA vs. DTT are compared, LA would have to be compared with the oxidized form of DTT, i.e. trans-4-5- dihydroxy-1,2-dithiane. The authors may discuss this issue, e.g. in the Discussion.

DHLA is a unique physiological low molecular weight dithiol. In this reduced form it is unstable, so most of researchers in their studies use its oxidized form, i.e. LA. It is well known that after administration to the biological systems, LA is transported to the intracellular compartments and is reduced to DHLA in reaction catalyzed by NAD(P)H-dependent enzymes. However, the degree of this reduction depends of the cell redox state. We believe that in the cell there is a kind of balance between LA and DHLA and the biological effect of administrated LA is the sum of its both redox forms. That is why we decided to study the effect of LA not only DHLA.

-In the Introduction, the authors wrote that "Furthermore, DSF-like side effects can occur when EtOH is used together with some plant-derived pharmaceuticals or dietary supplements which are often used without physicians supervision". In my opinion, specific examples of plant-derived pharmaceuticals or dietary supplements should be given here, as in the next sentence, where the authors present coprine, as an example of compounds which interact with EtOH to produce DSF-like reaction.

Appropriate examples and references have been added.

-I suggest you re-edit the Materials and Methods section. From the reader's point of view, this part is sometimes not very clear and chaotic. Although the reviewer understood this, it took him too much time. It should be corrected.

The section Materials and Methods section has been modified.

 -In the Results, the authors wrote: The obtained results demonstrated that the presence of 1 mM thiols (LA and DHLA) had no effect on the activity of intact ALDH. This sentence should be rewritten because LA is not a thiol. It is a disulfide.

It has been corrected.

 The obtained Results showed that DHLA was able to protect ALDH against the inhibitory effect of DSF (Fig. 1B). The ALDH activity changes here to 134.5% compared to the control (100%). Can the authors explain this? It would be good to discuss this problem in the Discussion.    

It was connected with the fact that commercially available enzyme contains some oxidized sulfhydryl groups. DHLA acts as a reductant and in this way increases total ALDH activity. LA as a disulfide had no effect on the activity of ALDH in this experiment. Similar results indicating an increase in ALDH activity under the influence of various concentrations of DHLA were obtained by us earlier (data presented during XLV Polish Annual Conference on Catalysis - 2013). This information has been discussed.

 -I also found a few language errors: typos, punctuation, etc. This should be corrected. For example: page 7 (167) - there are commas instead of dots; page 5 (128), 8 (182) and the following pages: the "in vitro" expression should be written in italics; page 8 (204) and 9 (231) - use the same abbreviation: either Me or 2-Me

All these type of errors have been corrected.

Round 2

Reviewer 2 Report

Intriguingly, results were replicated on mammalian model which increase chances it is translatable to humans. I also found interesting the fact that GSH is less effective toward ALDH protection than DHLA in vitro. GSH concentration is 1000-fold high in cell and it would be interesting to evaluate how this excess benefits its effectiveness compared to DHLA in the author'sfuture research. 

Overall, authors fully addressed all the comments, therefore, I recommend this paper for publication.